# Efficient Sim-to-real Transfer of Contact-Rich Manipulation Skills with Online Admittance Residual Learning

**Xiang Zhang**[*], **Changhao Wang**,[*]**Lingfeng Sun, Zheng Wu, Xinghao Zhu, Masayoshi Tomizuka**
Department of Mechanical Engineering
University of California at Berkeley, United States

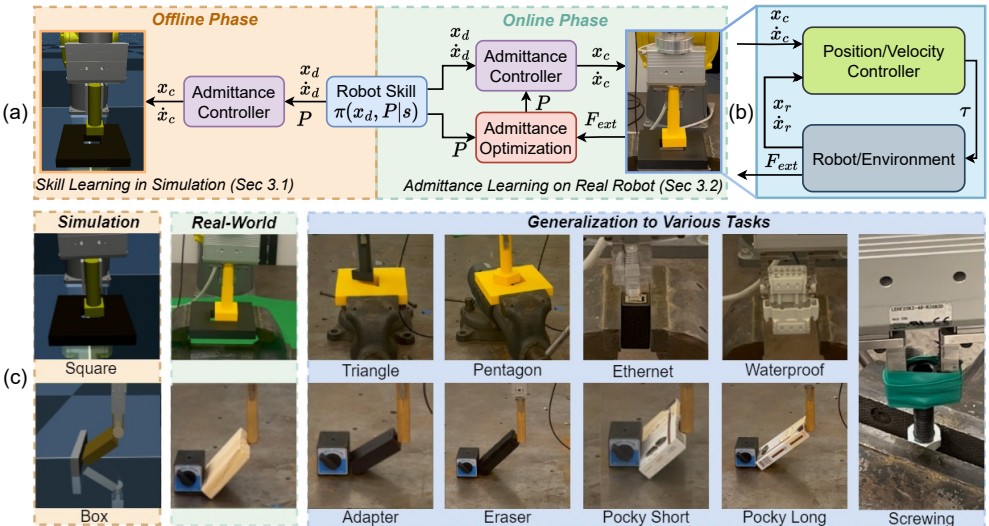

Figure 1: As shown in (a), we propose a robust contact-rich manipulation skill learning framework that offline learns the robot motion and compliance control parameters in the simulation and online adapts to the real world. The structure of the admittance controller is depicted in (b). Our framework demonstrates robustness in sim-to-real transfer and generalizability to diverse real-world tasks in (c)

**Abstract:** Learning contact-rich manipulation skills is essential. Such skills require the robots to interact with the environment with feasible manipulation trajectories and suitable compliance control parameters to enable safe and stable contact. However, learning these skills is challenging due to data inefficiency in the real world and the sim-to-real gap in simulation. In this paper, we introduce a hybrid offline-online framework to learn robust manipulation skills. We employ model-free reinforcement learning for the offline phase to obtain the robot motion and compliance control parameters in simulation with domain randomization. Subsequently, in the online phase, we learn the residual of the compliance control parameters to maximize robot performance-related criteria with force sensor measurements in real time. To demonstrate the effectiveness and robustness of our approach, we provide comparative results against existing methods for assembly, pivoting, and screwing tasks. Videos are available at https://sites.google.com/view/admitlearn.

**Keywords:** Contact-rich Manipulation, Admittance Control, Sim-to-real Transfer

---

[*]Equal Contribution. Correspondence: {xiang_zhang_98, changhaowang}@berkeley.edu

7th Conference on Robot Learning (CoRL 2023), Atlanta, USA.

# 1 Introduction

Contact-rich manipulation is common in a wide range of robotic applications, including assembly [1, 2, 3, 4, 5, 6, 7, 8], object pivoting [9, 10, 11], grasping[12, 13, 14], and pushing [15, 16]. To accomplish these tasks, robots need to learn both the manipulation trajectory and the force control parameters. The manipulation trajectory guides the robot toward completing the task while physically engaging with the environment, whereas the force control parameters regulate the contact force. Incorrect control parameters can lead to oscillations and excessive contact forces that may damage the robot or the environment.

Past works have tackled the contact-rich skill-learning problem in different ways. First, the majority of previous works [2, 10, 6, 7, 3, 17, 9, 18, 19] focus on learning the manipulation trajectories and rely on human experts to manually tune force control parameters. While this simplification has demonstrated remarkable performance in many applications, letting human labor tune control parameters is still inconvenient. Furthermore, the tuned parameter for one task may not generalize well to other task settings with different kinematic or dynamic properties. For example, assembly tasks with different clearances will require different control parameters. Another line of work deals with this problem by jointly learning the robot's motion and force control parameters [20, 21, 22, 23, 24, 25, 4, 26, 8]. Such learning processes can be conducted in both real-world and simulation. However, learning such skills on real robots is time-consuming and may damage the robot or environment. Learning in simulation is efficient and safe, however, the learned control parameters may be difficult to transfer to real robots due to the sim-to-real gap, and directly deploying the learned control parameters may cause damage to the robot.

In this paper, we focus on transferring robotic manipulation skills. We notice that the manipulation trajectory is more related to the kinematic properties, such as size and shape, which have a smaller sim-to-real gap and can be transferred directly, as demonstrated by previous works [9, 2, 10, 3]. However, simulating the contact dynamics proves to be challenging, primarily due to its sensitivity to various parameters, including surface stiffness and friction [27]. This sensitivity will result in a large sim-to-real gap and affects the learned compliance control parameters. Inspired by the above analysis, we propose a framework to learn robot manipulation skills that can transfer to the real world. As depicted in Fig. 1(a), the framework contains two phases: skill learning in simulation and admittance adaptation on the real robot. We use model-free RL [28, 29] to learn the robot's motion with domain randomization to enhance the robustness for direct transfer. The compliance control parameters are learned at the same time and serve as an initialization to online admittance learning. During online execution, we iteratively learn the residual of the admittance control parameters by optimizing the future robot trajectory smoothness and task completion criterion. We conduct real-world experiments on three typical contact-rich manipulation tasks: assembly, pivoting, and screwing. Our proposed framework achieves efficient transfer from simulation to the real world. Furthermore, it shows excellent generalization ability in tasks with different kinematic or dynamic properties, as shown in Fig. 1(c). Comparison and ablation studies are provided to demonstrate the effectiveness of the framework.

# 2 Related Works

## 2.1 Sim-to-real Transfer in Robot Contact-Rich Manipulation

Contact-rich manipulation tasks involve the interaction between robots and the environment through physical contact. In recent years, there has been a growing trend in utilizing simulation environments such as MuJoCo [30], Bullet [31], and IsaacGym [17] to learn and train robots for these tasks. These simulation environments offer advantages in terms of safety, scalability, and cost-effectiveness. Nevertheless, the sim-to-real gap remains a significant challenge.To address the gap, various approaches have been explored, including system identification, transfer learning, domain randomization, and online adaptation. System identification approaches [32, 33] involves the calibration of simulation parameters to improve accuracy and align the simulation with real-world dynamics. Transfer learn-

ing methods [34, 35, 25] aim to fine-tune skills learned in simulation for application in real-world scenarios. Domain randomization techniques [10, 9, 25, 36] are employed to create diverse environments with varying properties, enabling the learning of robust skills for better generalization. Instead of collecting large datasets in the real world, online adaptation methods [37, 38, 39, 40, 41, 42] utilize real sensor measurements to optimize a residual policy/model or directly update the policy network in real-time. Tang et al. [43] further improves the sim-to-real transfer performance by combining the above techniques with a modified objective function design for insertion tasks.

## 2.2 Learning Variable Impedance/Admittance Control

Compliance control [44], such as impedance and admittance control, enables robots to behave as a mass-spring-damping system. Tuning the compliance control parameters is crucial for stabilizing the robot and accomplishing manipulation tasks. However, manual tuning can be time-consuming. To address this issue, learning-based approaches have been applied to automatically learn the control parameters. Previous methods have focused on learning compliance control parameters either from expert demonstrations [20, 21, 22] or through reinforcement learning (RL) [23, 24, 25, 4, 26, 45] to acquire gain-changing policies. [21, 4, 20, 23, 25] propose to directly collect data in the real world. However, it is time-consuming to collect the data. Authors in [22, 26] have demonstrated success in directly transferring the learned control parameters from the simulation to the real world. Nevertheless, their applications are limited to simple tasks, such as waypoint tracking and whiteboard wiping.

## 3 Proposed Approach

We focus on learning robust contact-rich manipulation skills that can achieve efficient sim-to-real transfer. We define the skill as $\pi(x_d, P|s)$, which generates both the robot desired trajectory $x_d$ and the compliance control parameters $P$ given the current state $s$.

We use *Cartesian space admittance control* as the compliance controller. As shown in Fig. 1(b), the admittance control takes in the desired trajectory $[x_d, \dot{x}_d, \ddot{x}_d] \in \mathbb{R}^{18}$, and the external force/torque $F_{ext} \in \mathbb{R}^6$ measured on the robot end-effector and outputs the compliance trajectory $[x_c, \dot{x}_c, \ddot{x}_c] \in \mathbb{R}^{18}$ to the position/velocity controller according to the mass-spring-damping dynamics [44]:

$$M(\ddot{x}_c - \ddot{x}_d) + D(\dot{x}_c - \dot{x}_d) + K(x_c - x_d) = F_{ext} \tag{1}$$

where $M, K, D$ are the robot inertia, stiffness, and damping matrices, respectively. We assume $M, K, D$ is diagonal for simplicity and $P = \{M, K, D\}$ as the collection of all control parameters.

To achieve this goal, We propose an offline-online framework for learning contact-rich manipulation skills as depicted in Fig. 1(a). In the offline phase, we employ the model-free RL with domain randomization to learn the robot motion and the initial guess of compliance control parameters from the simulation (Section 3.1). In the online phase, we execute the offline-learned motions on the real robot and learn the residual compliance control parameters by optimizing the future robot trajectory smoothness and task completion criteria(Section 3.2).

### 3.1 Learning offline contact-rich manipulation skills

We utilize model-free RL to learn contact-rich manipulation skills in MuJoCo simulation [30]. The problem is modeled as a Markov decision process $\{S, A, R, P, \gamma\}$ where $S$ is the state space, $A$ is the action space, $R$ is a reward function, $P$ denotes the state-transition probability, and $\gamma$ is the discount factor. For each timestep $t$, the agent is at the state $s_t \in S$, executes an action $a_t \in A$, and receives a scalar reward $r_t$. The next state is computed by the transition probability $p(s_{t+1}|s_t, a_t)$. Our goal is to learn a policy $\pi(a|s)$ that maximizes the expected future return $\mathbb{E}\left[\sum_t \gamma^t r_t\right]$.

Specifically, we focus on learning robot skills for three contact-rich tasks: assembly, pivoting, and screwing. In these tasks, the robot needs to utilize the contact to either align the peg and hole

or continuously push and pivot the object, which makes them suitable testbeds for our proposed framework. The detailed task setups can be found below:

**Assembly Task:** The goal is to align the peg with the hole and then insert it.

*State space*: The state space $s \in \mathbb{R}^{18}$ contains peg pose $s_p \in \mathbb{R}^6$ (position and Euler angles) relative to the hole, peg velocity $v_p \in \mathbb{R}^6$, and the external force measured on the robot wrist $F_{ext} \in \mathbb{R}^6$.

*Action space*: The action $a \in \mathbb{R}^{12}$ consists of the end-effector velocity command $v_d \in \mathbb{R}^6$ and the diagonal elements of the stiffness matrix $k \in \mathbb{R}^6$. To simplify the training, the robot inertia $M$ is fixed to $diag(1, 1, 1, 0.1, 0.1, 0.1)$, and the damping matrix $D = diag(d_1, \cdots, d_6)$ is computed according to the critical damping condition $d_i = 2\sqrt{m_i k_i}, i = \{1, 2, 3, 4, 5, 6\}$.

*Reward function*: The reward function is defined as $r(s) = 10^{(1-\|s_{pos}-s_{pos}^d\|_2)}$, where $s_{pos} \in \mathbb{R}^3$ is the peg position and $s_{pos}^d \in \mathbb{R}^3$ is the nominal hole location. The exponential function encourages successful insertion by providing a high reward.

**Pivoting Task:** The goal is to gradually push the object to a stand-up pose against the wall.

*State space*: The state $s \in \mathbb{R}^{12}$ consists of the robot pose $s_p \in \mathbb{R}^6$ and the external force $F_{ext} \in \mathbb{R}^6$.

*Action space*: For simplicity, we consider a 2d pivoting problem: the robot can only move in the $X, Z$ direction. The robot action $a \in R^4$ contains the velocity command in $X, Z$ direction and the corresponding stiffness parameter.

*Reward function*: We use the rotational distance between the goal orientation $R^{goal}$ and current object orientation $R$ as the cost and define the reward function as $r = \frac{\pi}{2} - d$, with $d = \arccos\left(0.5(\text{Tr}(R^{goal} R^T) - 1)\right)$, which computes the distance of two rotation matrices between $R$ and $R^{goal}$. The constant term $\frac{\pi}{2}$ simply shifts the initial reward to 0. This reward encourages the robot to push the object to the stand-up orientation.

We use domain randomization on the robot's initial pose and contact force to improve the robustness of the learned skills. The implementation details can be found in Appendix. B.1.

### 3.2 Online Optimization-Based Admittance Learning

We have learned a policy that can perform contact-rich manipulation tasks in simulation. However, the sim-to-real gap may prevent us from directly transferring the learned skills to the real world. Our goal is to adapt the offline learned skills, especially the admittance control parameters, with online data in real time. Instead of retraining skills with real-world data, we propose locally updating the control parameters using the latest contact force measurements during online execution. We formulate online learning as an optimization problem that optimizes the residual control parameters to achieve smooth trajectory and task completion criteria while respecting the interaction dynamics between the robot and the environment. We will describe the optimization constraints, objective function, and overall online learning algorithm in Section 3.2.1, 3.2.2, and 3.2.3, respectively.

#### 3.2.1 Optimization Constraints

**Robot dynamics constraint:** Admittance control enables robot to behave as a mass-spring-damping system as shown in Eq. 1. We consider the robot state $x = [e, \dot{e}]$, where $e = x_c - x_d$, and we can obtain the robot dynamics constraint in the state space form:

$$\dot{x} = \begin{bmatrix} \dot{e} \\ \ddot{e} \end{bmatrix} = f(x, F_{ext}, u) = \begin{bmatrix} \dot{e} \\ -M^{-1}D\dot{e} - M^{-1}Ke + M^{-1}F_{ext} \end{bmatrix} \tag{2}$$

where the optimization variable $u = [m_1^{-1}, \ldots, m_6^{-1}, k_1', \ldots, k_6', d_1', \ldots, d_6']^T$ is the diagonal elements of $M^{-1}$, $K' = M^{-1}K$, $D' = M^{-1}D$. $e, \dot{e}$ are the robot states that can be directly accessed, and $F_{ext}$ is the external force that should be modeled from the environment dynamics.

**Contact force estimation:** Modeling the contact force explicitly is difficult because the contact point and mode can change dramatically during manipulation. Therefore, we propose to estimate

the contact force online using the force/torque sensor measurements. In our experiments, we utilize a simple but effective *record & replay* strategy, where we record a sequence of force information $\{F_{ext}^0, \ldots, F_{ext}^T\}$ within a time window $[0, T]$ and replay them during the optimization.

There are other approaches for force estimation, such as using analytical contact models [46, 47] or numerically learning the contact force by model fitting. However, we found the *record & replay* strategy is better by experiments. We provide analyses in the Appendix. C.2.

**Stability constraint:** To ensure stability for admittance control, we need the admittance parameters to be positive-definite. Therefore, we constrain the optimization variable $u$ to be positive.

### 3.2.2 Objective Function Design

We want to optimize the admittance parameters to establish stable contact and successfully achieve the task. Previous work [48] introduces the FITAVE objective $\int_0^T t|\dot{e}(t)|dt$ to effectively generate smooth and stable contact by regulating the robot's future velocity error. In addition, the ITAE objective $\int_0^{+\infty} t|e(t)|dt$ in [49] minimizes the position error to ensure the robot tracking the desired trajectory and finishes the task. We combine those two functions as our objective:

$$C(x) = \int_0^T t[w|e(t)| + (1-w)|\dot{e}(t)|]dt \tag{3}$$

where $w \in \mathbb{R}$ is a weight scalar to balance the trajectory smoothness and task completion criterion.

### 3.2.3 Online admittance learning

The optimization formulation is shown in Eq. 4. We optimize the residual admittance parameters $\delta u$, with $u_{init}$ obtained from the offline learned skill.

$$\begin{aligned} \min_{\delta u} \quad & C(x) \\ \text{s.t.} \quad & \dot{x} = f(x, F_{ext}, u_{init} + \delta u) \\ & F_{ext} \leftarrow \textit{record \& replay} \\ & u_{init} + \delta u > 0 \end{aligned} \tag{4}$$

We illustrate the online admittance learning procedures in Alg. 1. In the online phase, we execute the skill learned offline on the real robot and recorded the contact force at each time step. Every $T$ seconds, the online optimization uses the recorded force measurements, the current robot state and the admittance parameters learned offline to update the admittance parameter residual. The process runs in a closed-loop manner to complete the desired task robustly.

$$u = u_{init} + \delta u^*, M = diag\{m_1, \cdots, m_6\}, K = M \cdot diag\{k'_1, \cdots, k'_6\}, D = M \cdot diag\{d\prime_1, \cdots, d'_6\} \tag{5}$$

---

**Algorithm 1:** Online Admittance Residual Learning

---

**Require:** $u_{init}$ from the offline policy $\pi(a|s)$, current robot state $x$
1: **while** task not terminated **do**
2:    **if** every $T$ seconds **then**
3:       $\delta u^* \leftarrow$ admittance optimization in (4)
4:       $M, K, D \leftarrow$ Recover admittance parameters from (5)
5:    **end if**
6:    $\{F_{ext}\} \leftarrow$ record force sensor data
7: **end while**

---

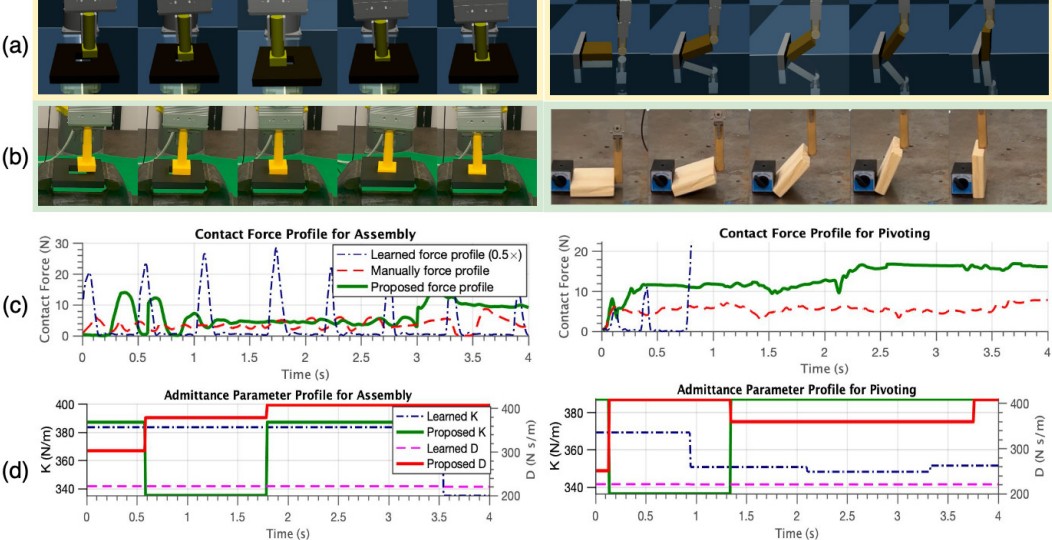

Figure 2: (a) shows the snapshots of the learned policy in simulation.(b) demonstrates the snapshots using the proposed approach for sim-to-real transfer. (c)(d) illustrate the forces and control parameters profiles for both the learned and proposed approach in the real world. The proposed approach can adjust the parameters to get the best performance in real-time.

## 4    Experiment Results

We conduct experiments on three contact-rich manipulation tasks, peg-in-hole assembly, pivoting, and screwing, to evaluate: 1) the robustness of sim-to-real transfer and 2) the generalizability of different task settings. We provide comparison results with two baselines in the assembly and pivoting tasks: 1) **Direct Transfer**: directly sim-to-real transfer both the learned robot trajectory and the control parameters [26], 2) **Manual Tune**: transfer learned trajectory with manually tuned control parameters [2]. We consider three metrics for evaluation: 1) **success rate** indicates the robustness of transfer, 2) **completion time** for successful trials denotes the efficiency of the skills, and 3) **max contact force** shows the safety. The screwing experiments further demonstrate the robustness of our method for solving complex manipulation tasks.

### 4.1    Skill Learning in Simulation

We use Soft Actor-Critic [28] to learn manipulation skills in simulation.[2]  During the evaluation, the learned assembly and pivoting skills both achieved a $100\%$ success rate. Fig. 2(a) shows the snapshots of the learned assembly skills. The robot learns to search for the exact hole location on the hole surface with a learned variable admittance policy and smoothly inserts the peg into the hole. For the learned pivoting skill, the robot pushes the object against the wall and gradually pivots it to the target pose with suitable frictional force.

### 4.2    Sim-to-Real Transfer

We evaluate the sim-to-real transfer performance on the same task. In the real world, the task setup, such as the object and robot geometry, is identical to the simulation. We mainly focus on evaluating the effect of the sim-to-real gap on robot/environment dynamics.

---

[2]We also tested other RL algorithms like DDPG [50] and TD3 [28]. As a result, all the methods are able to learn a policy and have similar performance when transferring to the real robot. Details can be found on our website.

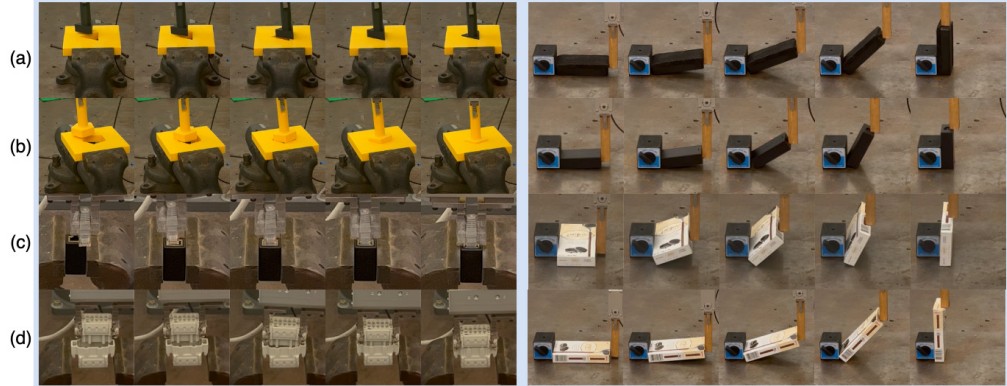

Figure 3: Snapshots of using the proposed approach to generalize to various task settings. The snapshots and videos of the baseline methods are available on our website.

We first apply the offline learned skill to the real world. From the experiments, we notice that the *Direct Transfer* baseline fails to produce safe and stable interactions. As depicted in Fig 2(c), for peg-in-hole assembly tasks, the peg bounces on the hole surface and generates large contact forces, making the assembly task almost impossible to complete. Similarly, in the pivoting task, the robot cannot make stable contact with the object and provide enough frictional force for pivoting.

Then we examine whether the learned robot motion is valid with manually tuned control parameters. As shown in Tab. 1, the *Manual Tune* baseline can achieve a $100\%$ success rate for both tasks. This supports our hypothesis and previous works that the manipulation trajectory is directly transferable with suitable control parameters to address the sim-to-real gap.

However, manual tuning requires extensive human labor. We want to evaluate whether the proposed online admittance learning framework can perform similarly without any tuning. Table 1 presents an overview of the sim-to-real transfer results. For all experiments, the weight parameter $w$ of the proposed approach was consistently set to $0.4$. Notably, our proposed method achieves a $100\%$ success rate in the assembly task, along with a $90\%$ success rate in the pivoting task. Furthermore, it achieves these results while exhibiting shorter completion times than the other two baselines.

We also investigate the contact force and the adjusted admittance parameters during the manipulation, shown in Fig. 2(c)(d). Initially, the robot establishes contact with the environment using the offline learned parameters, resulting in a large applied force. In the subsequent update cycle, the proposed method effectively adjusts the parameters by decreasing $K$ and increasing $D$, enabling the robot to interact smoothly with the environment and reduce the contact force. Later, it increases $K$ and decreases $D$ to suitable values to finish the task more efficiently.

| | Assembly Task | | | Pivoting Task | | |
|---|---|---|---|---|---|---|
| | Succ. Rate | Time (s) | Max F (N) | Succ. Rate | Time (s) | Max F (N) |
| Proposed | $10/10$ | $19.0 \pm 11.2$ | $23.6 \pm 6.3$ | $9/10$ | $25.6 \pm 2.1$ | $20.1 \pm 4.1$ |
| Manual | $10/10$ | $28.1 \pm 8.6$ | $10.3 \pm 2.2$ | $10/10$ | $25.3 \pm 3.6$ | $9.2 \pm 0.6$ |
| Direct | $3/10$ | $39.0 \pm 12.8$ | $63.7 \pm 6.8$ | $0/10$ | $N/A$ | $30.7 \pm 4.6$ |

Table 1: Success rate evaluation in real-world experiments.

## 4.3 Generalization to Different Task Settings

The aforementioned experiments highlight the ability of the proposed framework to achieve sim-to-real transfer within the same task setting. In this section, we aim to explore the generalization capabilities of the proposed approach across different task settings, which may involve distinct kinematic and dynamic properties. For two baselines, we directly use the manually tuned or learned control parameters of the training object for new tasks.

| | Triangle (gap = 1mm) | | Pentagon (gap = 1mm) | | Ethernet (gap = 0.17mm) | | Waterproof (gap = 0.21mm) | |
|---|---|---|---|---|---|---|---|---|
| | Succ. Rate | Time (s) | Succ. Rate | Time (s) | Succ. Rate | Time (s) | Succ. Rate | Time (s) |
| Proposed | 10/10 | $15.9 \pm 6.2$ | 10/10 | $20.1 \pm 8.9$ | 9/10 | $42.1 \pm 13.7$ | 9/10 | $37.8 \pm 17.7$ |
| Manual | 8/10 | $43. \pm 17.0$ | 9/10 | $38.0 \pm 18.0$ | 1/10 | $78.0 \pm 0.0$ | 0/10 | $N/A$ |
| Direct | 0/10 | $N/A$ | 1/10 | $7.0 \pm 0.0$ | 0/10 | $N/A$ | 0/10 | $N/A$ |

| | Adapter [L=8.8 cm, w=69g] | | Eraser [L=12.2 cm, w=36g] | | Pocky Short [L=7.9 cm, w=76g] | | Pocky Long [L=14.8 cm, w=76g] | |
|---|---|---|---|---|---|---|---|---|
| | Succ. Rate | Time (s) | Succ. Rate | Time (s) | Succ. Rate | Time (s) | Succ. Rate | Time (s) |
| Proposed | 8/10 | $25.0 \pm 4.8$ | 9/10 | $28.4 \pm 2.7$ | 8/10 | $12.9 \pm 1.7$ | 7/10 | $31.8 \pm 11.0$ |
| Manual | 0/10 | $N/A$ | 10/10 | $30.0 \pm 1.0$ | 1/10 | $19.0 \pm 0.0$ | 1/10 | $40.0 \pm 0.0$ |
| Direct | 0/10 | $N/A$ | 0/10 | $N/A$ | 0/10 | $N/A$ | 0/10 | $N/A$ |

Table 2: Generalization performance to different assembly tasks (**Top**) and pivoting tasks (**Below**).

**Peg-in-hole assembly:** We test various assembly tasks, including polygon-shaped peg-holes such as triangles and pentagons, as well as real-world socket connectors like Ethernet and waterproof connectors. These tasks are visualized in Fig. 1(c). The outcomes of our experiments are outlined in Table 2. Our proposed method achieves $100\%$ success rates on the polygon shapes and a commendable $90\%$ success rate on Ethernet and waterproof connectors. Moreover, the completion time of the proposed method is much shorter than other baselines. The *Manual Tune* baseline also achieves decent success rates on the polygon shapes as it is similar to the scenario in that we tune the parameters. However, for the socket connectors, due their tighter fit and irregular shapes, substantial force is required for insertion (approximately 15N for Ethernet and 40N for waterproof connectors) and *Manual Tune* baseline cannot accomplish these two tasks.

**Pivoting:** Similarly, we conduct a series of pivoting experiments on various objects, encompassing diverse geometries and weights as shown in Table 2. Remarkably, our proposed approach exhibits robust generalization capabilities across all tasks, achieving a success rate exceeding $70\%$. However, when relying solely on manually tuned parameters, the ability to pivot an object is limited to the eraser that has a similar length to the trained object and is the lightest object in the test set. As the object geometry and weight diverge significantly, the manually tuned parameters often fail to establish stable contact with the object and exert sufficient force to initiate successful pivoting.

**Screwing:** We conducted experiments on a more challenging robot screwing task to further validate our method. Its primary challenge is to precisely align the bolt with a nut and then smoothly secure them together. To address this, we employed the assembly skills previously learned for aligning the bolt and nut and then used a manually-designed rotation primitive to complete the screwing. Throughout the process, online admittance learning continually optimizes the admittance controller. Impressively, our approach allowed the robot to consistently and reliably align and secure the nut and bolt. We executed this task five times, achieving a $100\%$ success rate. The detailed settings can be found in Appendix. D.2 and the experiment videos are available on our website.

## 5   Conclusion and Limitations

This paper proposes a contact-rich skill-learning framework for sim-to-real transfer. It consists of two main components: skill learning in simulation during the offline phase and admittance learning on the real robot during online execution. These components work together to enable the robot to acquire the necessary skills in simulation and optimize admittance control parameters for safe and stable interactions with the real-world environment. We evaluate the performance of our framework in three contact-rich manipulation tasks: assembly, pivoting, and screwing. Our approach achieves promising success rates in both tasks and demonstrates great generalizability across various tasks.

However, there are some limitations of our proposed framework: 1) Our method refines the policy during execution, which means that initially, a sub-optimal policy is used to make contact with the environment. As a result, this scheme may not be suitable for contact with fragile objects. 2) We assume a simplified problem setup where the object is pre-grasped. However, real-world tasks may require the robots to first pick the object and then do the following manipulation tasks [43]. 3) We currently online learn/optimize the diagonal elements of admittance parameters. We'd like to consider learning other elements as suggested in [45].

**Acknowledgments**

We gratefully acknowledge reviewers for the valuable feedback, and we extend our thanks to the FANUC Advanced Research Laboratory for their insightful discussions on robot hardware and control.

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

# Appendices

## A   Task Setup

### A.1   Simulation Details

We build the simulation environment using MuJoCo [30] simulation for learning robot contact-rich manipulation skills. In the simulation, we include the model of the FANUC LRMate 200iD robot, and each of the joints is controlled with motor torque command. We incorporated an F/T sensor asset on the robot's wrist to measure the contact force. For the low-level controller, we employed computed torque control [51] to track the compliant trajectory $x_c$ and $\dot{x}_c$ derived from the admittance controller. The simulation time step was set to $0.01\ s$. Further details regarding the assembly and pivoting setups are outlined below:

**Assembly:** This task involves aligning a square-shaped peg with a hole. The edge length of the peg is $4\ cm$, and there is a clearance of $2\ mm$ between the peg and hole. The friction coefficient between the peg and hole is configured as $0.3$.

**Pivoting:** In this task, the objective is to reorient a rectangular object against a rigid wall. The simulated object has dimensions of $10 \times 10 \times 2.6\ cm^3$. A friction coefficient of $0.7$ is assigned to all objects in the simulation.

### A.2   Real Robot Experiment Setup

The real robot setup is visualized in Fig. 4. We utilized FANUC LRMate 200iD industrial robot as the test bed for our real-world experiments. The end-effector pose, and velocity are obtained from the joint encoders. The end-effector pose, and velocity are obtained from forward kinematics. The contact force is measured by an ATI Mini45 Force/Torque sensor mounted on the robot's wrist. The low-level position/velocity controller is achieved via a Positional-Integral (PI) control law with feed-forward terms to cancel gravity and friction. The controller is implemented in Matlab Simulink Real-Time and runs on $1KHz$. The admittance controller we use takes in the desired robot motion $x_d$ and optimized admittance control parameters $P$ and outputs the command robot motion $x_c$ to the low-level position/velocity controller. The robot motion $x_d$ is directly sent from an Ubuntu computer with a User Datagram Protocol(UDP) in $125Hz$. Similarly, the initial control parameters $P$ are sent from the Ubuntu computer and optimized in MATLAB with a built-in SQP solver.

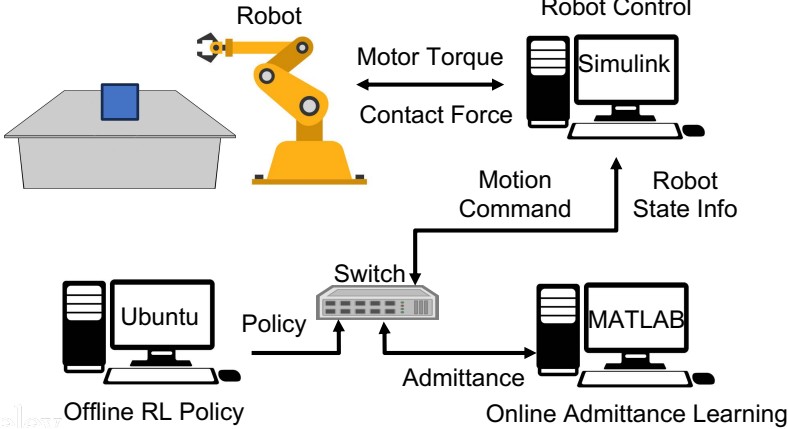

Figure 4: Real robot experiment setup.

# B   Simulation Training Details

## B.1   Domain Randomization Details for Contact-rich Tasks

In both the assembly and pivoting tasks, we introduced Gaussian noise with a mean of zero and a standard deviation of $0.2\ N$ to the FT sensor readings as measurement noise. Additionally, we applied a clipping operation to the collected contact force, limiting it to the range of $\pm 10\ N$ for regulation purposes. To enhance the robustness of the learned skills, we incorporated randomization into the robot's initial pose.

For the assembly task, the robot's initial pose was uniformly sampled from a range of $[\pm 30\ mm, \pm 30\ mm, 30 \pm 5\ mm]$ along the $X$, $Y$, and $Z$ axes, respectively. As for the pivoting task, the range for the initial pose was set to $[150 \pm 30\ mm, 5 \pm 5\ mm]$ along the $X$ and $Z$ axes relative to the rigid wall.

## B.2   RL Training Details

We use the Soft Actor Critic [28] with implementation in RLkit [52] to learn robot manipulation skills in simulation. The hyperparameter selections are summarized in Table. 3.

| Hyperparameters | Assembly | Pivoting |
|---|---|---|
| Learning rate - Policy | 1e-3 | 1e-4 |
| Learning rate - Q function | 1e-4 | 3e-4 |
| Networks | [128,128] MLP | [128,128] MLP |
| Batch size | 4096 | 4096 |
| Soft target update ($\tau$) | 5e-3 | 5e-3 |
| Discount factor ($\gamma$) | 0.95 | 0.9 |
| Replay buffer size | 1e6 | 1e6 |
| max path length | 20 | 40 |
| eval steps per epoch | 100 | 400 |
| expl steps per epoch | 500 | 2000 |

Table 3: Hyperparameters for RL training

# C   Discussion on Proposed Approach

## C.1   Discussion on the Necessity of Learning the Compliance Control Parameters

We consider the manipulation policy for contact-rich manipulation tasks to contain a manipulation trajectory and the corresponding compliance control parameters.

The main difference between 'contact-rich' manipulation and regular manipulation tasks is how much force the robot exerts on the environment. The more force the robot applies, the more force it has to withstand. For contact-rich manipulation, the robot desired trajectory often has to penetrate the object with its end-effector to generate enough force for the task. For example, to wipe a table, the robot has to push its end-effector below the table surface. Since the robot is a rigid object, it needs a compliance controller to regulate its behavior and prevent potential damage. Compared to a position/velocity controller that might not need to tune the PID gains frequently, a compliance control is very sensitive [48] to the change of environment or task goals. It thus requires careful tuning of the parameters for each task. Therefore, for contact-rich manipulation, a suitable policy should be matched with the appropriate compliance control parameters to achieve the task smoothly.

## C.2   Discussion on Approaches for Modeling Contact Force

A key component in our online admittance learning is the dynamics constraint, as shown below:

$$\dot{x} = \begin{bmatrix} \dot{e} \\ \ddot{e} \end{bmatrix} = f(x, F_{ext}, u) = \begin{bmatrix} \dot{e} \\ -M^{-1}D\dot{e} - M^{-1}Ke + M^{-1}F_{ext} \end{bmatrix} \tag{6}$$

where we want to regulate the future robot behavior based on the current robot state and the external force $F_{ext}$. In optimization, when we change the admittance parameter $M$, $K$, and $D$, the robot motion will change, and the external force that the environment gives to the robot will change as well. Thus, a robust way to model the external force $F_{ext}$ is crucial in our online admittance learning.

To estimate or approximate the contact force in real time, we compare four approaches:

- *record & replay*: We record the force/torque from the most recent measurements within a time window and directly use the pre-recorded data as $F_{ext}$ in the optimization.

- *hybrid impulse dynamics*: We use Eq. 6 with $F_{ext} = 0$ when there is no contact. For the contact, we model it implicitly as $M\dot{x}^- = \gamma M\dot{x}^+$, where $\dot{x}^-$ and $\dot{x}^+$ are the robot end-effector velocities before and after the contact. By online fitting the $\gamma$, we can optimize these hybrid dynamics to calculate the optimal parameters.

- *analytical contact model with online parameter fitting*: We model the contact explicitly using analytical models and fit the necessary parameters using online data, following [46, 47].

- *contact force fitting*: We fit a contact force model using online force sensor measurements.

However, the *hybrid impulse dynamics* approach is not suitable for our requirements. As shown in Fig. 5, the contact force profile in contact-rich manipulation indicates that the robot maintains contact with the environment for most of the time. Therefore, neglecting the entire contact process and modeling it implicitly is not appropriate for our applications.

Similarly, *analytical contact model with online parameter fitting* does not fit our scenarios either. Although it has been successful in some pivoting tasks, it relies on the quasi-static assumption that does not hold in our scenario. One of the main challenges of transferring the admittance parameters is to avoid the robot bouncing on the object. Moreover, the analytical model assumes point or sliding contact modes, which may be hard to generalize to different tasks, such as assembly.

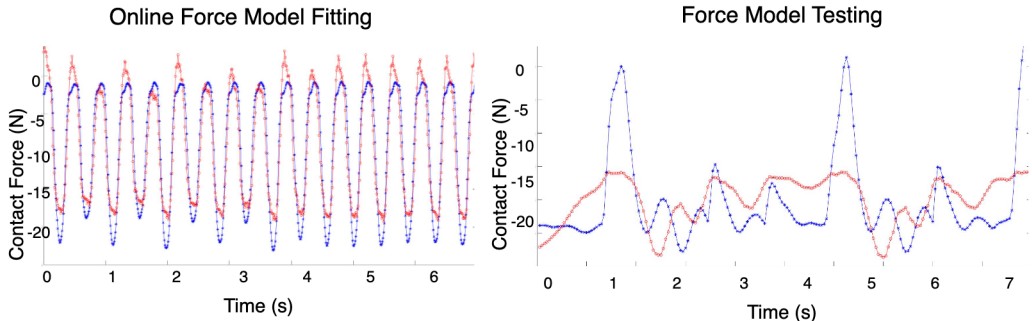

Figure 5: Performance of online force fitting (in $z$ axis). In every time window, we collect the force/torque measurements and use the least square to fit the force model $F_{ext}(x, \dot{x}) = a(t)x(t) + b(t)\dot{x}(t) + c(t)$. On the left, it shows the linear model can fit the force profile locally. However, it can be extremely challenging to generalize to the next time window, as shown on the right.

Finally, for *contact force fitting*, we assume a linear (spring-damping) contact force model: $F_{ext} = a(t)x(t) + b(t)\dot{x}(t) + c(t)$ within a short time window. We use the least square to estimate the parameters $a$, $b$, and $c$ in real-time. Fig. 5 shows an example of fitting results. It can fit the force profile well in a short time window. However, as we need to apply the model learned in the previous time window to the next step, the generalization ability is poor as it is hard to capture the peak of the force profile. Experiment videos comparing the performance of *contact force fitting* and *record & replay* are available on our website. We can observe that the contact force fitting method cannot stabilize the robot during contact.

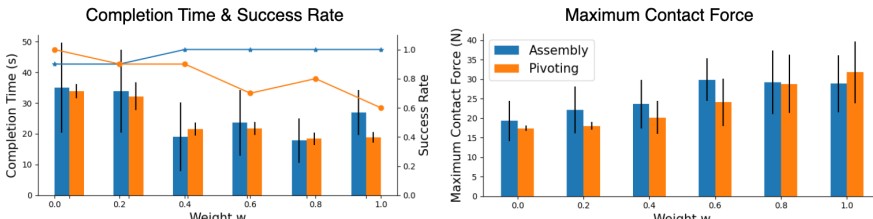

Figure 6: Ablation on the weight parameter. The left figure shows the completion time and success rate with respect to different w, and the right figure shows the contact force.

### C.3 Ablation: Objective Weight Selection

In this subsection, we would like to study the effect of weight selection. We evaluated different weight parameters on both the assembly and pivoting tasks. The results are depicted in Figure 6. For the assembly task, all proposed method variations achieve a $100\%$ success rate except for $w <= 0.2$. Smaller weight parameters tend to prioritize trajectory smoothness, which may not provide sufficient contact force for successful insertion. On the other hand, in pivoting tasks, larger weight values led to a decrease in the success rate. It is because larger weight values prioritize task completion, potentially leading to a failure in establishing a stable initial contact for pivoting. These observations align with the objective design motivation. Based on our findings, selecting the parameter $0.4$ strikes a good balance between both objectives and yields the best overall performance.

## D Baseline Results

### D.1 Sim-to-real Transfer

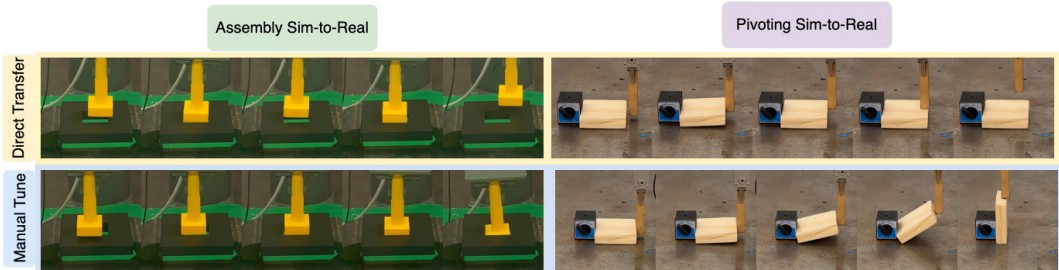

Figure 7: Snapshots of baseline approaches for the sim-to-real experiment. The control parameters learned in the simulation will result in a large contact force and makes the robot bounce on the surface, which will, in turn, result in failures of the tasks.

Here we provide snapshots of the baseline methods: *Direct Transfer* and *Manual Tune*. As introduced in the paper, *Direct Transfer* baseline utilizes the offline learned policy and directly applies it to the real robot without fine-tuning as [26] did. We hope the domain randomization on object position and force information can provide good generalizability and make it robust and transferable to real robots.

However, as shown in Fig. 7, direct applying the learned policy cannot achieve both tasks successfully. The main problem comes from the learned admittance control parameters. Where in the simulation, applying such parameters to the robot will not result in the robot bouncing on the object. In contrast, it can enable the robot to finish the task very efficiently. However, in the real world, such control parameters will result in large contact force and oscillation behaviors of the robot, which in turn, let the robot fails to establish stable contact with the object and finish the task.

For the *Manual Tune* baseline, we carefully tune the admittance parameters for each task in order to make the robot achieve smooth behavior during the contact. Table. 4 summarizes the parameters.

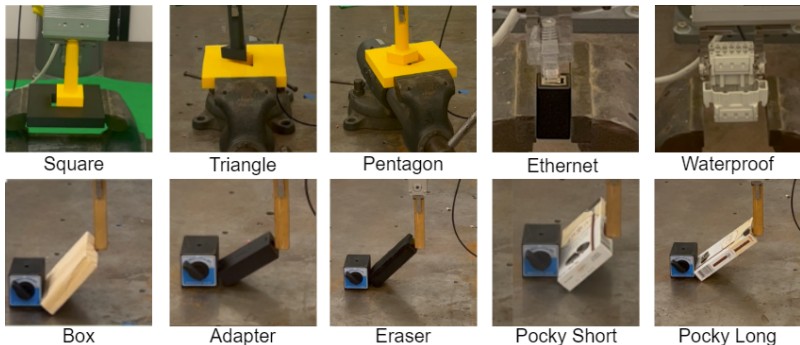

| Square | Triangle | Pentagon | Ethernet | Waterproof |

| Box | Adapter | Eraser | Pocky Short | Pocky Long |

Figure 8: Real-world manipulation tasks

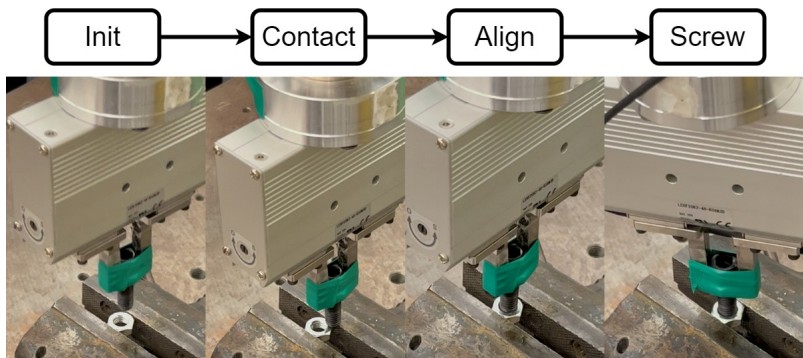

Figure 9: Snapshots of the screwing task

As shown in Fig. 7, the manually tuned baseline can successfully achieve the task. However, since it requires human tuning, it is not time-consuming and task-dependent. A practical problem of manually tuning the control parameters is the need of trying various combinations of parameters. During this process, it is dangerous to let the robot interact with the environment and may cause damage to both the object and the robot.

| Tuned Admittance Parameters | Assembly | Pivoting |
|---|---|---|
| End-effector Mass $M$ $(kg)$ | $[3, 3, 3]$ | $[4, 4, 4]$ |
| End-effector Inertia $I$ $(kgm^2)$ | $[2, 2, 2]$ | $[2, 2, 2]$ |
| Position Stiffness $K$ $(N/m)$ | $[200, 200, 200]$ | $[300, 300, 300]$ |
| Orientation Stiffness $K$ $(Nm/rad)$ | $[200, 200, 200]$ | $[200, 200, 200]$ |
| Position Damping $D$ $(Ns/m)$ | $[300, 300, 300]$ | $[300, 300, 300]$ |
| Orientation Damping $D$ $(Nms/rad)$ | $[250, 250, 250]$ | $[250, 250, 250]$ |

Table 4: Manually tuned admittance control parameters for the experiments.

## D.2 Robot Screwing Task

For the robot screwing task, we first execute the assembly policy that is learned previously for $8$ steps to align the bolt and nut. Then, we continuously apply a rotational motion which rotates the bolt by $20°$ in the yaw direction while pushing down the bolt to screw the bolt to the nut. We conducted experiments on an M8 bolt-nut assembly task for five times achieving a $100\%$ success rate. The experiment videos are available on our website. The snapshots of the screwing task are depicted in Fig. 9.

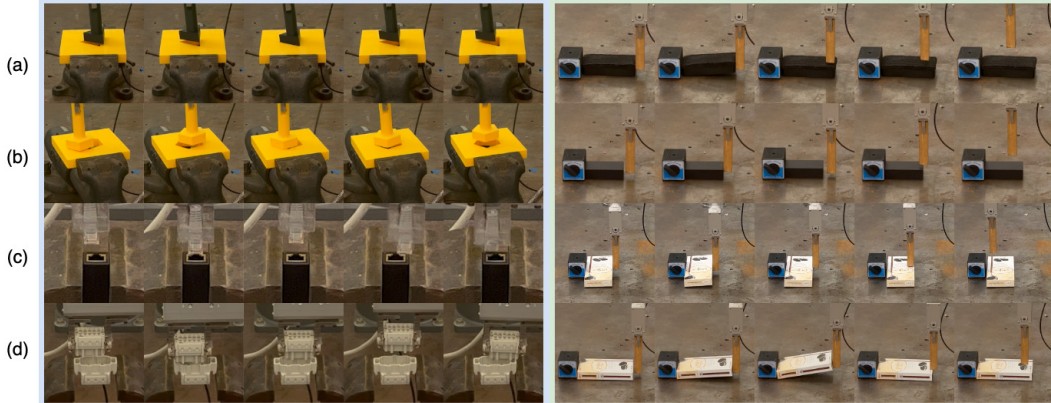

Figure 10: Snapshots of directly using the learned policy to generalize to various task settings. The snapshots and videos of the baseline methods are available on our website.

### D.3  Generalization to Different Task Settings

In order to evaluate the generalization performance to different tasks, we conducted tests on various variations of tasks as depicted in Fig. 8. For assembly, these tasks included polygon-shaped peg holes, such as triangular peg-holes with an edge size of $51.4\ mm$ and a clearance of $1.4\ mm$, as well as pentagon peg-holes with an edge size of $57.8\ mm$ and a clearance of $1.3\ mm$. Additionally, we performed experiments on standard electric connectors, such as Ethernet and waterproof connectors, for further assessment.

Regarding the pivoting task, we expanded the test set to include different objects. These objects consisted of an adapter with dimensions of $8.8 * 4.1 * 2.6\ cm^3$ and a weight of $69\ g$, an eraser with dimensions of $12.2 * 4.8 * 3.0\ cm^3$ and a weight of $36\ g$, and a pocky with dimensions of $14.8 * 7.9 * 2.3\ cm^3$ and a weight of $76\ g$.

The snapshots of the *Direct Transfer* and *Manual Tune* baselines can be seen in Fig.10 and 11, respectively. As observed in the sim-to-real experiments, the *Direct Transfer* baseline struggles to achieve stability during manipulation, resulting in failures when attempting to assemble or pivot objects of different shapes. On the other hand, the *Manual Tune* baseline demonstrates high success rates when dealing with polygon-shaped peg-holes and when pivoting the eraser. This success can be attributed to the similarity in geometric or dynamic properties between the learned object and these specific test objects. However, the *Manual Tune* baseline fails to generalize its performance to objects with significant differences, as illustrated in Fig.11(c) and (d).

## E  Current Limitations and Future Improvements

As we discussed in the paper, our current framework has three main limitations:

It assumes that the task settings in geometry are similar from training to testing. It uses a simple strategy for estimating the contact force. It has a relatively low update frequency and may not be suitable for manipulating fragile objects.

To address the first limitation, we plan to use meta-learning to learn the manipulation trajectory that can generalize well to different task settings. Meta-learning has been shown to be effective in generalizing the learned trajectory to various scenarios, and we believe that combining meta-learning and our proposed online residual admittance learning can bridge the sim-to-real gap for many contact-rich manipulation tasks. Safe reinforcement learning [53, 54, 55] is another domain that we'd like to explore, as it can provide safety guarantees of the learned policy to enable a safer and smoother initial policy.

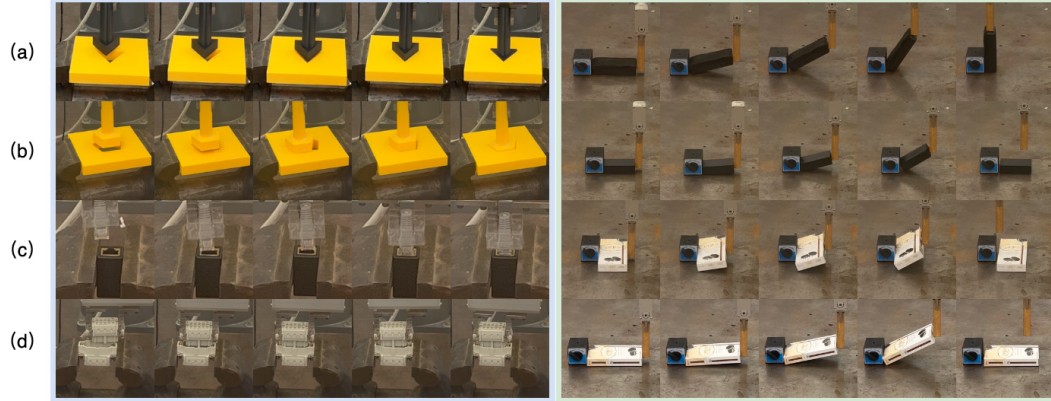

Figure 11: Snapshots of directly using learned trajectory and the manually tuned admittance control parameters to generalize to various task settings. The snapshots and videos of the baseline methods are available on our website.

For the second limitation, we are interested in exploring and experimenting with the analytical contact model approach as discussed in the Appendix. Using an analytical model and estimating the key parameters online may improve the performance. However, finding a general contact model or a method that can switch between different models will be the focus of our future work.

The last limitation is related to the time window size for online force/torque sensor data collection. We will try different time window sizes and increase the update frequency to enhance the adaptation performance in our future work. We also plan to incorporate recent advances in optimization to enable faster computation efficiency [56, 57].

