# OpenReview forum: "Efficient Sim-to-real Transfer of Contact-Rich Manipulation Skills with Online Admittance Residual Learning"
_robot-learning.org/CoRL/2023/Conference — CoRL 2023 Poster_

### Official Review · Reviewer_xCBP · 2023-07-20

**Confidence:** 4
**Originality:** Very Good
**Technical Quality:** Very Good
**Clarity Of Presentation:** Good
**Impact:** 3

**Recommendation:**

Strong Accept: I recommend accepting the paper and will argue for my recommendation even if other reviewers hold a different opinion.

**Review:**

This paper proposes an intuitive approach to addressing the sim-to-real gap: train a policy in simulation using standard methods, then use a lightweight online procedure to adapt the behavior to the real-world. This approach is appealing because (1) we can rely on existing RL methods that have been shown to be robust across many tasks, and (2) it offers a simple alternative to expensive procedures like SysID or Dynamics Randomization.

A key strength of this paper is the large number of real-robot results and thorough analyses that support the successful sim-to-real claims:
1. The main real-world results and videos show that the proposed method leads to a significant improvement over a baseline approach that does no sim-to-real adaptation.
2. Additional ablations analyze the importance of the weight parameter, “w”, for sim-to-real transfer.
3. Finally, more real-robot results test the policy’s ability to generalize to other geometries.

The main weaknesses of this paper are missing details that would help the reader better understand and apply the proposed method. Below is a list of additional details/questions:
- Additional details about the online optimization procedure. What value of T is used in the experiments? How long does it take to run the optimization (i.e., is it real-time or does it interrupt policy execution)?  Is the method sensitive to the window size used?
- Generalization experiment details. In these experiments, objects are used with different kinematic and geometric properties. Given that the proposed method mainly focuses on adapting the admittance parameters (and not the trajectory, x_d), could the authors please provide insight into why the proposed method succeeds even when the geometry differs significantly from that used in training?
- Gains clarification. For online admittance learning, I found it unclear exactly how the policy, u_init, and du interacted. For example, in Equation 4, is u_init a matrix of size Tx6 representing the gains output by the policy at each time step? Or a single vector of the gains at the beginning of the episode? Similarly, is du a single value for each gain dimension or is it also time varying? Expressing the domains of each variable would help clarify this point.
- MuJoCo Simulator: Was any SysID performed (even heuristically) on any simulation parameters that the authors found important to get reliable sim-to-real transfer?
- Figure 4: I believe the lines correspond to Success Rate and the bars to Completion Time. It would be helpful to clarify this in the caption.

I believe addressing the above points would add clarity and further bolster the strong results presented in this work.

----- Post-Rebuttal Update -----

I would like to thank the authors for their detailed answers to my questions. The added details, in the rebuttals to each reviewer, clarify a lot and would all be valuable in the final paper. In particular, the details about the action parameterization and control strategies. I have updated my recommendation to a strong accept.

My remaining question that I believe can be addressed in the final version of the paper is regarding the domain randomization. If I understand the rebuttal response, It appears as though pose randomization was sufficient to deal with objects of different sizes. This insight would be useful to discuss and include additional details about when this approach may fail.

**Quality Of The Limitations Section:**

Limitations are addressed clearly

**Questions For Rebuttal:**

- Online optimization clarifications (see above review)
- Gains clarifications (see above review)

**Robotics Focus:**

Sufficient demonstration on hardware

**Summary Of Paper:**

This paper focuses on the sim-to-real problem for contact-rich tasks like peg-in-hole insertion and object-pivoting. While kinematics generally transfer well from simulation, dynamics pose a bigger challenge as phenomena such as contact and joint-friction are harder to model accurately. The main contribution is a two stage framework for addressing the sim-to-real gap. In the first phase, a policy is trained in simulation using standard reinforcement learning methods (Soft Actor-Critic). The second phase addresses the sim-to-real gap by optimizing for admittance parameters online using an objective that encourages smooth motion that completes the task. The objective is a novel combination of previously proposed objective functions to minimize future position and velocity error. Via a set of real world experiments, the authors show the proposed framework greatly outperforms a method that has no sim-to-real procedure and is comparable to manually tuning the gains of the controller.

**Summary Of Recommendation:**

I weakly recommend accepting this paper. The paper presents an intuitive idea that is backed by strong results. The main weaknesses are missing details to help the reader fully understand the method. However, I believe these issues are addressable and look forward to the rebuttal period.

---

### Official Review · Reviewer_4kUn · 2023-07-20

**Confidence:** 5
**Originality:** Good
**Technical Quality:** Good
**Clarity Of Presentation:** Very Good
**Impact:** 4

**Recommendation:**

Weak Accept: I recommend accepting the paper, but will not argue for my recommendation if the majority of other reviewers have a different opinion.

**Review:**

The paper introduces a compelling hybrid offline-online framework for contact-rich manipulation skills, which is a noteworthy contribution to the field. The authors have done an excellent job in presenting their research with clear and organized writing. The incorporation of simulation and experimental results adds to the robustness of their findings.

However, my primary concern lies in the accurate estimation of contact forces during the manipulation process. As mentioned, discrepancies between simulation and real-world measurements, coupled with variations in the sampling rate of the real-world system for a residual estimator, raise questions about the feasibility of sim-to-real applications in this context. To strengthen the paper's significance, I believe a thorough analysis of these critical points is imperative, considering that it constitutes the main contribution of the research.

I suggest the authors delve deeper into addressing the challenges associated with sim-to-real transferability, providing a comprehensive comparison of simulation results against real-world data at various sampling rates. Furthermore, discussing potential mitigations or adaptations to bridge the gap between simulation and reality would greatly enhance the paper's impact and practical applicability.

Overall, the paper shows promise, but a more comprehensive analysis of the estimation of contact forces and sim-to-real transferability is essential for solidifying the research's credibility and practical implications.

**Quality Of The Limitations Section:**

Additional details required

**Questions For Rebuttal:**

Can you provide more insights into the specific methods used for contact force estimation in both the simulation and real-world experiments?
How do these methods differ, and what are the primary factors contributing to the observed discrepancies?

Considering the observed differences between simulation and real-world measurements, what strategies were employed to enhance the sim-to-real transferability in your hybrid framework?
Did you explore any techniques to narrow the sim-to-real gap in your experiments?

**Robotics Focus:**

Sufficient demonstration on hardware

**Summary Of Paper:**

The paper proposes a novel approach to address the challenges in learning contact-rich manipulation skills for robots. Such skills are crucial for safe and stable interactions with the environment, but they are difficult to acquire due to data inefficiency in the real world and the sim-to-real gap in simulations. To overcome these obstacles, the authors introduce a hybrid offline-online framework.
To validate their approach, the authors present comparative results against existing methods for assembly and pivoting tasks. The experimental results showcase the superiority of their hybrid offline-online framework in achieving contact-rich manipulation skills.
This methodology presents a significant advancement in robotic manipulation and has the potential to enable robots to learn and perform complex tasks safely and efficiently.

**Summary Of Recommendation:**

The paper presents a compelling hybrid offline-online framework for contact-rich manipulation skills, making a noteworthy contribution to the field. The research is well-written and organized, and the incorporation of simulation and experimental results adds robustness to the findings.

Considering the excellence demonstrated in this research, I believe that the paper will be a valuable addition to the scientific community and will contribute significantly to the advancement of robotic manipulation skills. I recommend accepting this paper for the CoRL.

---

### Official Review · Reviewer_zBMt · 2023-07-21

**Confidence:** 4
**Originality:** Good
**Technical Quality:** Fair
**Clarity Of Presentation:** Good
**Impact:** 3

**Recommendation:**

Weak Reject: I recommend rejecting the paper, but will not argue for my recommendation if the majority of other reviewers have a different opinion.

**Review:**

Strengths:
- The proposed method shows generalization capabilities for sim2real -- it significantly outperforms a non adaptive and a manually tuned baseline for 2 out of 4 objects for assembly and 3 out of of 4 objects for pivoting.
- The evaluation of a reward hyperparameter in Figure 4 is surprisingly uncommon -- and commendable

Major weaknesses:
- It's not clear to me why you learn the policy only offline and admittance only online -- did you try learning both offline or fine tuning both online? This seems like an important ablation to have
- Space between peg and hole is large -- 2 mm
- Maybe a personal view -- but if you're getting 100% success rate on self designed tasks in simulation (and later in sim2real) should you make the tasks harder? Otherwise it's not easy to see what are the limitations of the proposed method
- Figure 2 is important but unclear (see questions). Analysis of the results in the figure is also not really provided (Edit: it appears later in the paper -- should provide analysis when the figure is introduced)
- Comparisons to some relevant recent works are missing (see questions)
- Manual tuning does better than the proposed method in the sim2real experiments for insertion and pivoting (especially for F) -- the advantage stated of the proposed method is that manual tuning requires extensive human labor -- can you be more specific about your manual tuning process?
- In figures 3c and 3d it's hard to see what's going on. It looks like it's from the real world but there is an unusual background and artifacts that makes it seem like it's rendered. What are you using to grip the parts? And what kind of robot? It's really hard to see (Edit: it's much clearer in the website videos -- maybe the photos should be adjusted)
- There have been previous sim2real works for insertion with an online phase (such as a meta learning approach like Solowjow's work) but there don't seem to be any comparison to their methods
- No video attached to paper

Minor weaknesses:
- "to accomplish these tasks robots need to learn both the manipulation trajectory and the force control parameters" -- not exactly accurate -- there are papers that do contact rich tasks without learning control parameters and/or without using force control
- x_d is referred to as desired trajectory, but as far as I can tell, it's just a desired state
- Pivoting task has a simplified 2D action space
- There are small grammar issues throughout the paper

**Quality Of The Limitations Section:**

Limitations are addressed clearly

**Questions For Rebuttal:**

- "assembly tasks with different clearances will require different control parameters" -- could you give a specific example?
- Two recent papers that aren't mentioned are Kozlovsky et al "Reinforcement learning of impedance policies," and Tang et al "Industreal." The first does asymmetric admittance matrix learning and the second shows zero shot sim2real. Could you compare your methods/results to these works in detail?
- What is the reasoning behind the robot inertia matrix values?
- "exponential function encourages successful insertion by providing a high reward" -- did you compare to other rewards?
- How big are the peg and hole (what is the gap size)? Edit: see it in Appendix A.1 -- 2 mm -- this is a lot!
- I don't really understand the record & replay strategy. Do you optimize over a window but just freeze the force profile?
- Can you explain the t in the Fitave and Itae cost functions? Probably should explain it in the paper too
- Why does minimizing velocity error ensure smoothness? Doesn't it depend on whether the target velocity profile is smooth?
- Is equation 5 really needed? since you describe it earlier
- What made you choose SAC? (compared to PPO, DDPG, etc)
- Figure 2: what is the difference between the learned force profile and the proposed force profile? What does it mean that the learned profile has 0.5x next to it? Were all 3 profiles 100% successful? Why does the proposed force profile exceed the manual force profile for pivoting? What were the manual K and D values?
- line 185: "as depicted in Fig 2c" -- so is the "learned force profile" the same as the same as the "direct transfer"?
- When watching some of the videos online it seems that the robot takes an action, pauses, takes an action, pauses and so on -- why is that?

**Robotics Focus:**

Sufficient demonstration on hardware

**Summary Of Paper:**

The paper focuses on sim2real of contact rich manipulation skills for insertion and assembly. If I understood right there are 2 simulated envs (1 insertion and 1 assembly) and 8 real envs (4 insertion and 4 assembly). The key contribution is a sim2real method that learns controller targets for an admittance controller in simulation with some nominal admittance gains -- and then fine tunes the admittance gains from online optimization. This is an interesting idea but the simulation results were limited and the real world results (on known objects) were not better than a manual controller -- the one advantage seems to be in generalization to some of the unseen objects.

The paper has some weaknesses such as a lack of ablations, large spacing in the insertion task and a lack of comparison to some relevant recent works. It's clear that the online phase helps for certain objects but the proposed online method isn't compared to any others from the literature. See comments for more details.

**Summary Of Recommendation:**

The paper proposes an interesting online phase to improve sim2real transfer -- but it's difficult to know how effective the method is due to inconsistent results in simulation and sim2real, a lack of ablations, and a lack of comparisons to other online methods. The paper could be much stronger if more detailed experiments and comparisons were done and more thorough explanations were provided for cases where the proposed method does not show a significant advantage. Sim2real efforts all take a major amount of time and effort and the authors are commended for this line of work -- but this paper may need more time to develop so that it can have higher impact.

---

### Official Review · Reviewer_4sjw · 2023-07-21

**Confidence:** 3
**Originality:** Fair
**Technical Quality:** Good
**Clarity Of Presentation:** Very Good
**Impact:** 2

**Recommendation:**

Weak Accept: I recommend accepting the paper, but will not argue for my recommendation if the majority of other reviewers have a different opinion.

**Review:**

Quality: The presented work is of good quality, considering that the authors manage to deploy their method on a real robotic platform. Nonetheless, the quality of the work can be significantly improved by considering other baselines rather than only SAC (I believe this is quite a straightforward thing to do given that they used a toolkit).

Clarity: Both the main paper and the supplementary document are clear. However, I would encourage the authors to add more experimental detail e.g. the robot used, the toolkit, etc in the main document.

Originality: The work is somewhat original, albeit the use of a hybrid offline-online framework is not entirely new.

Significance of this work: This work could potentially contribute towards using more hybrid models in industrial assembly tasks.

Main Strengths: The main strength of this work is the fact that the method was deployed on a real industrial robot, which I appreciate can be quite challenging.

Main Weaknesses: The main weakness of this work is that the tasks chosen are quite simple and it would be ideal to motivate these and potentially consider other assembly tasks rather than only insertion.

**Quality Of The Limitations Section:**

Additional details required

**Questions For Rebuttal:**

(1) Why did you use specifically the MuJoCo simulator (you also mentioned other simulators)? Is this mainly because of its strength for contact tasks? (but again you did not model the contacts in the simulation).
(2) In 3.1 define MDP.
(3) Could you elaborate on the tasks chosen (pivoting, peg-in-hole) and why did you limit your experiments to this? It would be nice to see other assembly tasks e.g. screwing which is a challenging contact task as well.
(4) Why did you choose SAC and did you compare to other model-free baselines?

**Robotics Focus:**

Sufficient demonstration on hardware

**Summary Of Paper:**

The authors introduce a hybrid offline-online framework to learn robust manipulation skills, with a particular focus on assembly and pivoting tasks. Their method learns the skill in the simulation environment using MuJoCo and then adapts the admittance on the real robot. They carry out both simulation and real-world experiments.

**Summary Of Recommendation:**

I would recommend that following additional experiments (baselines, tasks), the paper could be accepted for publication.

---

### Author Response · Authors · 2023-08-10
**Summary of Additional Results**

We thank all the reviewers for their valuable comments and suggestions. We have prepared a detailed response and conducted additional simulations and real-world experiments to further support our paper:
- ablation on various model-free RL algorithms, including SAC, DDPG, and TD3
- ablation on reward function design for the peg-in-hole task (exponential reward and 2-norm distance reward)
- smaller clearance in simulation (0.5mm and 1mm) for peg-in-hole
- real robot experiments on the screwing task
- contact force comparison in simulation and real world

The simulation and real-world experiment videos are available on our website [https://sites.google.com/view/admitlearn](https://sites.google.com/view/admitlearn). Please let us know if further clarification is needed.

---

### Author Response · Authors · 2023-08-15
**Thanks for the Review. Please Let us Know if there are any other questions**

Dear reviewers,

We deeply appreciate your time in reviewing our paper! Your comments and suggestions help us a lot in improving the paper.

As it comes to the end of the rebuttal phase, we hope we have solved all your concerns and we are always ready to address any of your remaining concerns. Thanks again for taking the time to read our work and provide helpful feedback!

Paper Authors

---

### Decision · Program_Chairs · 2023-08-30

**Decision:**

Accept (Poster)

**Comment:**

The paper proposes a two-step approach to tackle the sim2real gap for contact-rich manipulation where a policy is first learnt in simulation using reinforcement learning, and in the second step an online compliance controller is optimized to obtain smooth motions. The approach is tested in real industrial robot for assembly and pivoting tasks, and compared to manual tuning the controller.

Reviewers were mostly positive about the work. They commented on usefulness of the main idea and the fact that it was tested on an industrial robot. The main concerns were related to specific details about the choices for the algorithms (eg SAC), lack of ablations, and simplicity of the tasks. These concerns were mostly addressed during the rebuttal phase with authors submitting a new version of the paper with significant more ablations and explanations.